# Thermal Insulation Foam of Polystyrene/Expanded Graphite Composite with Reduced Radiation and Conduction

**DOI:** 10.3390/polym17081040

**Published:** 2025-04-11

**Authors:** Pengjian Gong, Minh-Phuong Tran, Piyapong Buahom, Christophe Detrembleur, Jean-Michel Thomassin, Samuel Kenig, Quanbing Wang, Chul B. Park

**Affiliations:** 1Microcellular Plastics Manufacturing Laboratory (MPML), Department of Mechanical and Industrial Engineering, University of Toronto, 5 King’s College Road, Toronto, ON M5S 3G8, Canada; pgong@scu.edu.cn (P.G.); minh.phuong.tran.fr@gmail.com (M.-P.T.); piyapong@mie.utoronto.ca (P.B.); 2College of Polymer Science and Engineering, Sichuan University, 24 Yihuan Road, Nanyiduan, Chengdu 610065, China; 3Center of Education and Research on Macromolecules (CERM), CESAM Research Unit, Department of Chemistry, University of Liege, Allée de la Chimie, B6a, Sart Tilman, 4000 Liège, Belgium; christophe.detrembleur@ulg.ac.be (C.D.); jean-michel.thomassin@celabor.be (J.-M.T.); 4WEL Research Institute, 1300 Wavre, Belgium; 5Department of Polymers and Plastics Engineering, Shenkar College, Anne Frank Street 12, Ramat Gan 52526, Israel; samkenig@shenkar.ac.il; 6Jiangxi Tongyi Polymer Material Technology Co., Ltd., Jiangxi Tongyi New Material Industrial Park, Xinfeng County, Ganzhou City 341600, China; wangquanbing@tongyiplastic.com

**Keywords:** expanded graphite, IR absorption, IR extinction coefficient, thermal conductivity, insulation foams, supercritical carbon dioxide

## Abstract

Expanded graphite (EG) with high infrared (IR) absorption is incorporated at low concentrations (≤2 wt%) into polystyrene (PS) foams to reduce radiative thermal conductivity and solid thermal conductivity, which account for 20~40% and 10~30% of total thermal conductivity, respectively. After systematically and quantitatively investigating thermal insulation behavior in PS/EG foams, it was found that the inclusion of 1 wt% EG in 25-fold expanded PS/EG foam blocks over 90% of the radiative thermal conductivity, with only a marginal increase in heat conduction. A great reduction in total thermal conductivity from 36.5 to 30.2 mW·m^−1^·K^−1^ was then achieved. By further optimization using a co-blowing agent in the supercritical CO_2_ foaming process, superthermal insulating PS/EG foam with a total thermal conductivity of 19.6 mW·m^−1^·K^−1^ was achieved for the first time. This significant result implies that the composite material design together with the foaming process design is capable of obtaining a superthermal insulating composite foam by using the following strategy: using additives with high IR absorption efficiency, a foam with a large expansion ratio, and a co-blowing agent with low gas conductivity.

## 1. Introduction

Micro-/nanocellular composite foams, known for their excellent thermal insulating properties, are widely used in aerospace, automotive, electronics, and packaging applications [1,2,3]. Particularly in the construction industry, polymeric foam-based thermal insulating materials play a crucial role due to their low cost and high energy-saving potential. In recent years, extensive research has been conducted to expand the understanding of polymeric foams, enhance their thermal insulation performance, and elucidate the mechanisms of heat transfer through foam structure [1,2,3,4,5].

The heat transfer behavior of polymeric foams is typically governed by gas conduction, solid conduction, and thermal radiation [6]. For thermal transport in gas, the micro-/nanocells create confined spaces that restrict gas molecular movement, and natural convection becomes negligible. In sufficiently small confined spaces, collisions between gas molecules are reduced, and gas conduction is influenced by energy transfer between gas molecules and cell walls [7]. Solid conduction in polymeric foams can be modeled using the Glicksman model, particularly for low-density foams with a high void fraction (>94%) [6]. This model accounts for the influence of solids located in struts and cell walls on heat conduction through foam. At a given strut fraction, the overall heat conduction depends on the thermal conductivities of the gas and the solid, as foam with a higher expansion level has a larger fraction of the gas, which has a lower thermal conductivity than the polymer and, thereby, possesses lower overall heat conduction. With a larger strut fraction, heat conduction paths become more tortuous and solid conduction is hindered. Radiative heat transfer, on the other hand, involves the attenuation of thermal radiation due to multiple scattering and absorption events by struts and cell walls as electromagnetic waves pass through polymeric foams. Based on the well-established Rosseland equation, the radiative heat transfer at a given temperature is inversely proportional to the overall attenuation property of the material. This correlation describes the attenuation of thermal radiation with a spectrum of wavelengths within Planck’s spectral energy distribution for blackbody radiation. The Rosseland equation has become a crucial tool in modeling radiative heat transfer in microcellular foams [8,9,10,11]. Indicated by the so-called Rosseland mean extinction coefficient, the foam’s overall attenuation property depends on the individual radiation blocking feature of each cell wall and strut and their number densities in the foam structure. A marginal cell number density for insufficient attenuation events in overly large-sized foams or too-thin and highly transparent cell walls and struts in low-density foams could lead to an increase in the radiative thermal conductivity [12,13]. As the foam’s total thermal conductivity is influenced by gas conductivity, solid conductivity, and radiative thermal conductivity, both the foam structure and the material properties are crucial factors in the development of superior thermal insulating foams.

Among conventional non-vacuum insulation foams, highly expanded closed-cell polymeric foams are known to exhibit lower thermal conductivity than open-cell structures. However, the radiative heat transfer constitutes a significant portion (20–40%) of the total heat transfer in low-density polymeric foams, even with a closed-cell structure, and it increases further with an increased foam expansion level as cell walls become thinner and more transparent to thermal radiation [5,12,14]. Enhancing the infrared (IR) absorption feature of the foam matrix is an effective strategy to reduce thermal radiation in low-density foams. Recently, Wang et al. [12] and Buahom et al. [13,15] presented a model for radiative thermal transport through closed-cell polymeric foams, examining IR reflectance, absorbance, and transmittance. Their findings suggest that increasing the IR absorption index of the foam matrix can effectively block thermal radiation. Carbonaceous materials, known for their efficient IR absorption, are widely utilized to mitigate radiative heat transfer in thermal insulating materials [16]. For instance, surface-modified nano-graphite particulates [17], carbon black [14], graphite [18,19], carbon nanotubes (CNTs) [20,21], dispersed graphene fillers [22], and graphene oxide [23,24,25] could be incorporated in the polymer matrix to reduce its thermal radiation. The study of Gong et al. [5,20] demonstrated the effectiveness of CNTs in enhancing IR absorption in polystyrene (PS) foams with fine cell sizes (approximately 5 µm) and high expansion ratios (>18-fold). Their results showed that 75% of the radiative thermal conductivity was effectively blocked, reducing the radiative contribution to just 6.1% of the total thermal conductivity in PS/CNT foams with 1.0 wt% CNTs. Similar to CNTs, expanded graphite (EG) is a carbonaceous filler composed of parallel graphene sheets stacked along the c-axis [26]. EG as an eco-friendly material is derived from natural graphite, a widely available and sustainable resource. During usage, EG does not release any harmful gases or pollutants; after usage, EG can be reused and recycled. Such a characteristic of EG helps reduce environmental contamination, minimize waste generation, and support circular economy principles. With its strong IR absorption capacity [16,27,28], EG is expected to serve as an effective IR absorber when embedded in a PS matrix. Consequently, the addition of EG to foams is anticipated to reduce the radiative heat transfer and, by extension, the total thermal conductivity. Compared to CNTs, EG offers the advantages of lower cost and greater abundance, making it a promising candidate for the large-scale production of superthermal insulating materials.

In terms of environmental sustainability, the combination of high thermal insulation performance and the supercritical carbon dioxide (scCO_2_) foaming process offers significant advantages. These include environmental friendliness, easy recyclability, non-toxicity, and non-flammability [20,29,30,31,32]. Although foams containing conventional insulation gases such as chlorofluorocarbon (CFC) and hydrochlorofluorocarbon (HCFC) initially exhibit total thermal conductivity below 30 mW·m^−1^·K^−1^, the long-term permeation of these gases out of these foams, as highlighted by the Montreal Protocol in 1996, negatively impacts the ozone layer [33]. Compared with traditional insulation gases, trans-1-chloro-3,3,3-trifluoropropene (identified as S-propene) has a global warming potential (GWP) equal to that of CO_2_ (standardized to 1), does not deplete the ozone layer (similar to CO_2_), and breaks down quickly in the atmosphere with low long-term environmental impact. In addition, S-propene has no flammability under standard conditions. Therefore, this next-generation foam blowing agent complies with environmental regulations such as the Montreal Protocol. In this context, besides CO_2_ as a major physical blowing agent, S-propene can meet environmental requirements and serve as a co-blowing agent to enhance thermal insulation performance [34]. Therefore, our pioneering study focuses on the thermal conductivity of PS/EG foams prepared using the scCO_2_ and co-blowing agent foaming method.

In this work, the thermal conductivity of low-density PS/EG composite foams filled with various filler contents was studied. The radiative thermal conductivity was determined by calculating the extinction coefficient and applying the Rosseland equation [35,36]. The enhancement from adding EG to low-density foams to reduce the total effective thermal conductivity was systematically and quantitatively demonstrated using an analytical model that considers the Knudsen effect and heat conduction through the foams based on Glicksman’s model [6,20]. Compared to PS/CNT composites, the significantly higher IR absorption index of PS/EG composite foams could lead to a much stronger radiation blocking effect, resulting in a substantial reduction in the radiative heat transfer of PS/EG foams.

To achieve superthermal insulating polymeric foams with a total thermal conductivity (*λ_total_*) of less than 20 mW·m^−1^·K^−1^, it is desirable to further decrease the gas conductivity (*λ_gas_*) through the foams while maintaining high volume expansion (over 30-fold), good foam morphology with reduced solid conductivity (*λ_solid_*), and even dispersion of EG in the foams to reduce radiative conductivity (*λ_rad_*). By quantifying each heat transfer term (gas conduction, solid conduction, and radiation) of PS/EG foams, our work highlights the significant role of EG in reducing radiative thermal conductivity and describes gas and solid conduction through low-density PS/EG foams. The advanced scCO_2_ co-blowing agent foaming technology enabled us to achieve a total thermal conductivity of less than 20 mW·m^−1^·K^−1^ for the first time.

## 2. Materials and Methods

### 2.1. Materials

Polystyrene (PS Styrolution^®^ GPPS 3100—BASF, 1.04 g·cm^−3^ density), with a melt flow index (MFI) of 10 g/10 min (@ 200 °C) and a thermal conductivity of 180 mW·m^−1^·K^−1^, was received from the supplier. Grade H-5 expanded graphite (EG) with an average platelet diameter of 5 μm was supplied by XG-Science, East Lansing, MI, USA. N-methyl pyrrolidone (NMP) (>99.8%) C_5_H_9_NO was supplied by Sigma Aldrich, Darmstadt, Germany. A commercial solvent Chromasolv Plus^®^ was supplied from Honeywell, Michigan, USA. The polyol (diols) Acclaim 8200 (M_w_~8000 g·mol^−1^, based on polypropylene oxide) was purchased from Covestro, Leverkusen, Germany. Trans-1-Chloro-3,3,3-trifluoropropene (Solstice^®^ 1233zd(E), denoted as S-propene gas), with a boiling point of 19 °C, a liquid density of 1.27 g·cm^−3^, and a vapor thermal conductivity of 10.2 mW·m^−1^·K^−1^ at 20 °C, was supplied from Honeywell, NC, USA. Carbon dioxide with a purity of 99.8% was purchased from Linde Gas Canada, ON, Canada.

### 2.2. Preparation of PS/EG Composites

EG treatment: 4.0 g of EG was added to 200 mL of ethanol and magnetically stirred. The 2.0 g polyol (Acclaim^®^ 8200 diols), used as a surface treatment agent, was added into the black suspension of EG/ethanol; this was followed by an ultra-sonication process for 5 h at 40 °C. The surface-modified EG was placed in Petri dishes and then dried under ambient conditions for 2 days (in a fume hood). Before compounding, the surface-modified EG was placed in a vacuum oven for 3 days to remove the solvent residue.

PS/EG composites: Various EG contents composites were then prepared, containing 0.1 to 2.0 wt% of modified EG, by compounding the modified EG with neat PS using a DSM twin-screw compounder (Xplore MC15 15 mL) at 200 °C and 100 rpm for 5 min.

### 2.3. Foaming Process

The neat PS and PS/EG composite rod samples, 5 mm in diameter and 30 mm in length, were placed into a high-pressure autoclave for saturation with the physical co-blowing agent CO_2_–pentane or CO_2_–S-propene. A volume of 30 mL of liquid pentane (boiling point of 36.1 °C) or liquid S-propene (boiling point of 19 °C) was poured into the autoclave (163.0 mL chamber volume). CO_2_ was then pumped into the chamber until a pressure of 5.9–13.8 MPa was achieved, and the samples were soaked for 4 days at 70 °C. Foaming was induced by a rapid depressurization. Then, the foamed samples were quickly placed into 100 °C boiling water for 2.5 min for further expansion.

### 2.4. Thermal Conductivity Measurement

A transient plane source (TPS) Hot Disk Thermal Constants Analyser (Therm Test Inc., TPS 2500, Hanwell, NB, Canada) was used to measure the total thermal conductivity (*λ_total_*) of the solid bulk samples as well as foam samples. Hot Disk Thermal Analysis software version 7.3 records temperature increases in the surrounding sample and then determines the radiative thermal conductivity. In this study, the sample used for thermal conductivity measurement was about 15 mm in diameter. The measurement was performed three times for each sample at different positions, and the experimental deviation was less than 5% [37,38].

### 2.5. Scanning Electron Microscopy Characterization

Scanning Electron Microscopy (SEM) was used to observe the microstructure of the foam composites and the neat foams. The samples were fractured in liquid nitrogen and then coated with gold for 180 s before SEM (JEOL JSM-606, JEOL Inc., Tokyo, Japan) observations. Image-J processing software version 1.50 was used to calculate the cell density (*N_f_*). The samples’ bulk densities before and after foaming were measured using the water displacement method based on ASTM D792–00 [39]. The cell density was then calculated by the following equation [40]:(1)$$N_{f}=\frac{\rho_{s}}{\rho_{f}}{\left(\frac{n}{A}\right)}^{1.5}$$
where *n* is the number of cells in the total area, *A*. The *ρ_f_* and *ρ_s_* are the densities of the foam and the unfoamed bulk, respectively.

### 2.6. Transmission Electron Microscopy Characterization

Transmission Electron Microscopy (TEM) was used to observe the dispersion of the EG in the PS matrix. Samples (with 1.0 wt% EG) were cut into thin films of 150 nm thickness using a Leica Microtome EM UC6 at ambient temperature. TEM micrographs were obtained by using Philips CM100, 100 kV-Canon Tungsten, Philips Electron Optics, Netherlands.

### 2.7. Fourier-Transform Infrared Analysis

Fourier-Transform Infrared (FTIR) Spectroscopy (Perkin Elmer Spectrum One) was used to measure the spectral transmittance of radiation through the foams. The spectral transmittance was collected by averaging 8 scans in a spectral range of 4000–400 cm^−1^ (a wavelength of 2.5–25 μm) with a spectral resolution of 4 cm^−1^. Before recording the samples’ IR transmittance, the background noise caused by H_2_O, CO_2_, N_2_, Ar, etc., in the air was registered. The foams were cut into a disk shape of 15 mm in diameter, with thicknesses ranging from 0.2 to 2 mm. To calculate the radiative thermal conductivity, we measured at least 6 disk samples of different thicknesses.

The IR absorption index (*A_index_*) of the film samples was calculated using IR transmittance and reflectance through the thin films. The films’ internal transmittance (*τ_film_*) can be calculated as follows [41]:(2)$$\tau_{film}=\left(\frac{2Ref\left(2-Ref\right)}{C-\sqrt{C^{2}-4Ref\left(2-Ref\right)}}-1\right)\frac{1}{\tau }$$
where τ and Ref are the transmittance and reflectance, respectively. *C* is a function of both properties, defined as follows:(3)$$C=1+2Ref+\tau^{2}-Ref^{2}$$

By the linear regression of $\ln\tau_{film}$ against the film thickness $L_{film}$, the film’s IR absorption index (*A_index_*) was determined from the slope.(4)$$\ln\tau_{film}=-A_{index}\left(\frac{4\pi }{\lambda }\right)L_{film}$$
where *λ* is the thermal radiation wavelength.

## 3. Results and Discussion

### 3.1. Composite Morphology

Figure 1a shows SEM micrographs of the cryogenic fracture surface of the neat PS without fillers (smooth surface in SEM micrograph), while Figure 1b presents the fracture surface of the PS/EG 1.0 wt% composite with EG fillers (rough surface in SEM micrograph). A TEM image of the PS/EG 1.0 wt% composite is presented in Figure 1c. Because the dispersing state of EG in the composite foams plays an important role in blocking radiation, optical microscopy (OM) was used to demonstrate the dispersion of EG in the PS/EG 1.0 wt% foam sample, as shown in Figure 1d. The micrographs of the EG particles in the Appendix A show that the average platelet sizes of EG were less than 5 μm, which corresponds to Figure 1d.

### 3.2. Effect of EG on Radiative Thermal Conductivity in Composite Foams

Neat PS exhibits weak IR absorption within the 2.5–25 μm wavelength range. Therefore, carbonaceous fillers are commonly incorporated to enhance IR blocking. The intrinsic IR absorption index of the composites was calculated using Equations (2)–(4) based on the IR internal transmittance and sample film thickness. The addition of carbonaceous fillers such as EG led to a decrease in the IR transmission through the bulk solid PS composite while increasing the IR absorption index, as seen in Figure 2. The wavelength-averaged IR absorption index of PS/EG and PS/CNT composites at various filler contents are presented in Table 1. It is noted that compared with CNTs [5], the addition of EG significantly increased the IR absorption index and the derived IR blocking efficiency. For instance, the PS/EG 1.0 wt% composite showed a similar IR absorption index as the PS/CNT 2.0 wt% composite. This indicates that EG provides superior IR blocking performance compared to CNTs at the same filler concentration.

With respect to the foams, Figure 3a shows the spectral transmittance of the neat PS foam with a density of 0.06 g·cm^3^ at various sample thicknesses of 200, 400, 740, 990, and 1330 μm, respectively. Higher transmittance indicates greater radiative energy transfer through the samples. It is noted that the IR transmittance decreased with the sample thickness. This is because a thicker sample contains more cell walls, which reflect and absorb radiation before it reaches the FTIR detector [42]. The interaction between radiation and foam structure depends on the matrix composition, including the type and concentration of fillers. Figure 3b–f illustrate the IR transmittance of PS/EG foams containing 0.1–2.0 wt% EG, with a density of around 0.04 g·cm^−3^ at different sample thicknesses. Despite having a slightly lower density than neat PS foam, all PS/EG composite foams exhibited lower IR transmittance. This suggests that the reduction in IR radiation was due to the dispersed IR-absorbing EG particles embedded within the PS matrix (Figure 1d).

The FTIR spectral transmittance data for different foam thicknesses (Figure 3a–f) have been analyzed to determine the spectral extinction coefficient *K_e,λ_* at specific wavelengths, as seen in Figure 4. A higher *K_e,λ_* value (spectral extinction coefficient) indicates greater IR attenuation. The Rosseland mean extinction coefficient (*K_e,R_*) represents the average of *K_e,λ_* over the entire wavelength range from 2.5 to 25 μm, calculated using Appendix A. *K_e,R_* was then used to determine the radiative thermal conductivity (*λ_rad_*) by using Appendix A. Apparently, the addition of EG significantly increased the IR blocking property (indicated by the Rosseland mean extinction coefficient) due to the enhanced IR absorption of the foam matrix.

In order to elucidate the effect of the volume expansion ratio on the radiative heat transfer, foams with varying volume expansion ratios and different EG contents were fabricated by using either scCO_2_–pentane or scCO_2_–S-propene as co-blowing agents. As shown in Figure 5, increasing the filler content (EG and CNT) led to a reduction in the radiative heat transfer. It is also noted that the radiative thermal conductivity increased with a higher expansion ratio in both neat and filler-containing polymeric foams. The increased expansion ratio resulted in a decrease in cell wall thickness [43]. When the cell wall thickness becomes too thin, a high level of radiative energy passes through the foam without being attenuated [12,13]. In this case, the solid matrix with a strong IR absorption characteristic is preferred to compensate for the thin cell wall in highly expanded foams. The IR blocking efficiency of the PS/EG foams is illustrated in Figure 6. Significant absorption occurred even at low EG content. The IR absorption increased from 0% to 80% within the range of 0–0.5 wt% EG. Moreover, higher IR absorption occurred when a higher content of EG was used. For example, 1.0 wt% of EG resulted in 92% absorption, and 2.0 wt% of EG exhibited a 98% absorption, almost entirely blocking the radiation. It is very interesting to note that the PS/EG foams displayed lower radiative thermal conductivity than the PS/CNT foams at the same filler content and the same expansion ratio of foams (Figure 5 and Figure 6).

Figure 7 shows that the average radiative absorption efficiency of the PS/EG and PS/CNT foams (y-axis) is related to the wavelength-averaged IR absorption index (presented in Table 1) of the bulk PS/CNT and bulk PS/EG composites (x-axis) at different filler contents. It is noted that an increase in carbonaceous filler’s content enhanced the IR absorption index of the solid matrix and correspondingly increased the radiative absorption efficiency of the foams. The effect of the EG content on the reduction in IR radiation is more significant at higher foam expansion where the radiative thermal conductivity is significant for neat PS (Figure 5). Figure 7 shows that, at low filler contents of 0.1–0.5 wt% (EG and CNT), the average radiative absorption efficiency of the PS/EG foams increased from 0.61 to 0.82, which is higher than that of the PS/CNT foams (from 0.35 to 0.68). The results come from the fact that the wavelength-averaged IR absorption index in the bulk PS/EG composite was higher than that of the PS/CNT (Table 1). At higher filler contents of 1.0–2.0 wt%, average radiative absorption efficiencies as high as ~0.9 (for PS/EG foams) and ~0.8 (for PS/CNT foams) were achieved. Although the wavelength-averaged IR absorption index in the bulk PS/EG significantly increased when more EG was added (0.0025 at 1.0 wt% to 0.0035 at 2.0 wt%), the average radiative absorption efficiency of the PS/EG foams slightly increased from 0.89 to 0.92. This behavior was similar to that of the PS/CNT foams when its average radiative absorption efficiency increased from 0.79 to 0.84 for the PS/CNT 1.0 wt% and PS/CNT 2.0 wt% foams, respectively. Since EG is a lower-cost additive and has higher IR absorption efficiency than CNTs, it implies that the use of low-cost and high-IR-blocking-efficient EG is a more effective choice for superthermal insulating foam production than CNTs. The contents of EG and CNTs were kept around 2.0 wt% to optimize the performance of radiative absorption and the contribution of solid conduction.

### 3.3. Morphology of Composite Foams

Figure 8, Figure 9 and Figure 10 show the foam morphology of the neat PS and PS/EG foams with different EG content. The SEM micrographs demonstrate good uniformity in cell size and a polygonal cell shape. Figure 11 shows the cell size, the cell density, the expansion ratio, and the strut fraction of all samples foamed in the co-blowing agents scCO_2_–pentane and scCO_2_–S-propene.

### 3.4. Effect of EG on Total Thermal Conductivity in Composite Foams

A direct comparison between the *calculated* and *experimental* thermal conductivities of the PS/EG composite foams is presented in Figure 12a. In PS/EG foams, the values of the gas conductivity (*λ_gas_*) and the solid conductivity (*λ_solid_*) were marginally maintained at approximately 24 mW·m^−1^·K^−1^ and 4.5–6.5 mW·m^−1^·K^−1^, respectively. Hence, the decrease in the total thermal conductivity is primarily attributed to the major reduction in the radiative thermal conductivity when incorporating EG into the PS matrix. The EG’s effect as an IR absorber was more evident in the case of the PS/EG 1.0 wt% sample, where the radiative thermal conductivity (*λ_rad_*) decreased from 7.7 to 1.3 mW·m^−1^·K^−1^. This led to a lower *calculated* total thermal conductivity in the PS/EG 1.0 wt% foam (30.2 mW·m^−1^·K^−1^) compared with that of the neat PS foam (36.5 mW·m^−1^·K^−1^), as shown in Figure 12a. Compared with the neat PS foam, all the PS/EG composite foams exhibit lower total thermal conductivity due to the presence of EG fillers as IR absorbers. However, increasing the EG content beyond a certain level was not effective in further decreasing the total thermal conductivity. This is attributed to the high thermal conductivity of EG at high loadings contributing to an increase in the solid conductivity. In addition, too-high EG content could hinder the foaming process. Thus, an optimal level of EG should be incorporated into PS foam to balance the IR absorption and its inherent thermal conductivity.

As described in Figure 12a,b, the radiative contribution of EG to the total heat transfer decreased from 21% for neat foam to 4% at 1.0 wt% EG and nearly vanished to 2% at 2.0 wt% EG. The contribution of solid conduction (*λ_solid_*) was maintained within the range of 14–21% (PS/EG composite foams), while the gas conductivity (*λ_gas_*) was the dominant contribution accounting for 65–80% of the total thermal conductivity. Therefore, to achieve superthermal insulating polymeric foams (<20 mW·m^−1^·K^−1^), a further decrease in gas conductivity and thereby overall heat conduction is desirable in addition to maintaining high volume expansion (>30-fold), uniform foam morphology (to reduce solid conductivity), and EG dispersion in foams (to reduce radiative thermal conductivity, *λ_rad_*). Instead of pentane, S-propene was used as a co-blowing foaming agent (scCO_2_ and S-propene). The advantages of using S-propene are two-fold: (1) its low bulk gas conductivity (10.2 mW·m^−1^·K^−1^) and (2) the formation of a high expansion ratio foam caused by its strong plasticization effect on PS. The reduction in gas conductivity is highlighted when the PS/EG-B foam has the lowest total thermal conductivity of 19.6 mW·m^−1^·K^−1^ (Appendix A). This outstanding result is attributed to three factors: (1) a ~50-fold expansion ratio (reducing the solid conductivity); (2) effective blocking of IR radiation with the addition of EG (reducing the radiative thermal conductivity in foams with high volume expansion); and (3) the usage of the S-propene co-blowing agent (reducing the gas conductivity).

Most CO_2_ diffuses out of PS foam after 10 days, whereas around 70% of HCFC-142b diffuses out after 10,000 days (27 years) [19]. HCFC-142b gas is widely used as an insulation gas in PS foams, and it has nearly the same thermal conductivity and molecular weight as S-propene. The effective diffusivity of HCFC-142b is 6 × 10^−10^ (cm^2^·s^−1^), and we expected that the effective diffusivity of S-propene in polystyrene would not be far from this value. Therefore, the S-propene used in this work was expected to diffuse out as slow as HCFC-142b does and that the PS/EG-B foam would maintain ultra-low thermal conductivity for many years [19,44].

Finally, Figure 13 summarizes the effectiveness of carbonaceous fillers as IR absorbers in polymer foams. Accordingly, the total thermal conductivities of foams with different carbonaceous levels and foam expansion ratios as reported in the literature and investigated in the current study are compared. The foams were prepared by the physical blowing agents scCO_2_, N_2_, and n-pentane and/or by the chemical blowing agent azocarbonamide. As evident from Figure 13, the thermal conductivity decreased with increased carbonaceous filler content and there is an optimal expansion ratio. At low expansion ratios (<10-fold), composite foam demonstrated high thermal conductivities (higher than 40 mW·m^−1^·K^−1^) due to the large contribution of the solid conductivity (*λ_solid_*). An increase in the expansion ratio (5- to 10-fold) results in the total thermal conductivity’s decrease from 110 to 50 mW·m^−1^·K^−1^. Most of the foams based on PS and PE have a thermal conductivity of around 30–40 mW·m^−1^·K^−1^ at expansion ratios higher than 18-fold. In high expansion ratios, i.e., over 20-fold, gas conduction (*λ_gas_*) and radiation (*λ_rad_*) dominate the heat transfer of the foams. With respect to radiation, the carbonaceous additive demonstrated its high effectiveness in reducing the radiative thermal conductivity at large expansion ratios, and hence total thermal conductivity decreased from ~40 to ~30 mW·m^−1^·K^−1^. As gas conductivity makes a major contribution to heat transfer, the low-conductivity S-propene enhances the thermal insulation performance. Compared to the literature data, the investigated and developed PS/EG composite foams of the present study demonstrate the lowest thermal conductivity. Furthermore, the total thermal conductivity of 19.6 mW·m^−1^·K^−1^ is the lowest reported experimental value for polymeric foams made by scCO_2_-assisted foaming technology, as shown in Appendix A. In addition, recent studies on different materials for thermal insulation are also summarized in Table 2 to highlight the advantages of our low-density foams.

## 4. Conclusions

A detailed study was conducted on the effect of EG on the heat transfer properties of low-density PS foams. PS/EG composite foams were prepared using co-blowing agents, achieving high expansion ratios of approximately 20-fold (scCO_2_–pentane) and 35–50-fold (scCO_2_–S-propene). The contribution of each heat transfer mode to the total thermal conductivity was systematically and quantitatively determined based on models of gas conduction, solid conduction, and radiation. The effect of EG as an IR absorber was evident in the PS/EG foams, significantly reducing the radiative thermal conductivity from 7.7 mW·m^−1^·K^−1^ (neat PS foam) to 1.3 mW·m^−1^·K^−1^ (PS/EG 1.0 wt% foam) while maintaining similar foam morphology. This resulted in much lower total thermal conductivity in the PS/EG 1.0 wt% foam, at around 30 mW·m^−1^·K^−1^, compared to the neat PS foam at 36.5 mW·m^−1^·K^−1^. Considering EG as an eco-friendly additive and scCO_2_ foaming as a green processing method, the insulating PS/EG foams offer both energy-efficient and environmentally friendly features. The PS/EG composites could be manufactured using industrial twin-screw extrusion and the scCO_2_ foaming could be processed in industrial mold foaming technology, which implies the high potential for industrialization of the novel thermal insulating foam fabricated.

## Figures and Tables

**Figure 1 polymers-17-01040-f001:**
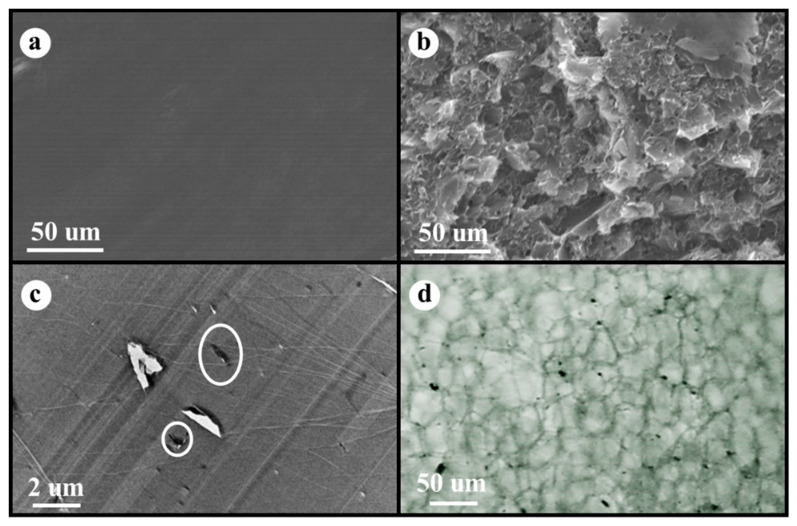
SEM micrographs of (**a**) neat PS and (**b**) PS/EG 1.0 wt%, (**c**) TEM image of PS/EG 1.0 wt% (white circle indicates EG), (**d**) OM micrograph of PS/EG 1.0 wt% foam (black dots are EG particles).

**Figure 2 polymers-17-01040-f002:**
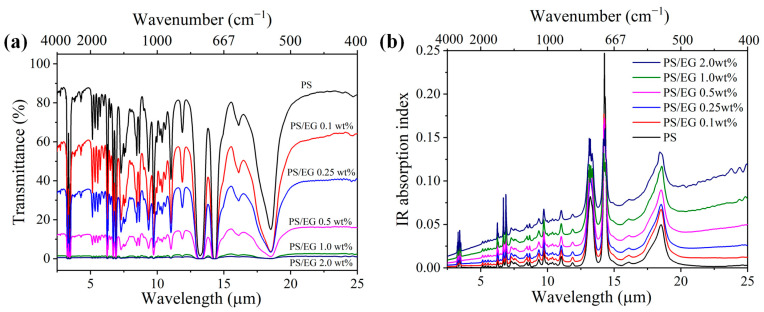
(**a**) FTIR transmittance spectra and (**b**) IR absorption index of neat PS and PS/EG composite films (~50 μm thick) in the wavelength range of 2.5–25 μm.

**Figure 3 polymers-17-01040-f003:**
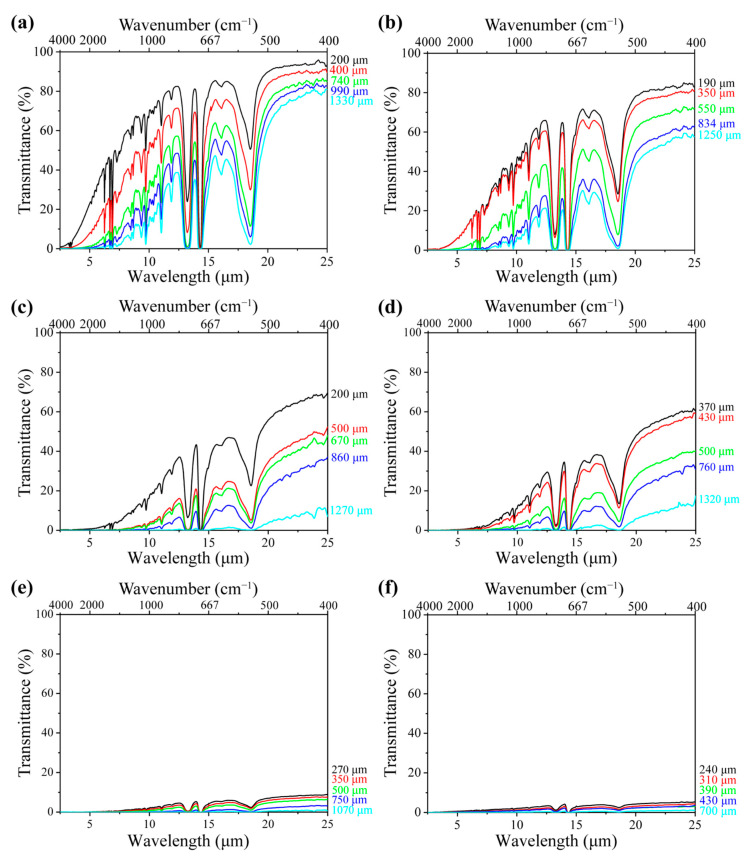
FTIR spectral transmittance of (**a**) neat PS foam; (**b**) PS/EG 0.1 wt% foam; (**c**) PS/EG 0.25 wt% foam; (**d**) PS/EG 0.5 wt% foam; (**e**) PS/EG 1.0 wt% foam; and (**f**) PS/EG 2.0 wt% foam (with varying foam thicknesses—samples foamed in CO_2_–pentane at 13.8 MPa).

**Figure 4 polymers-17-01040-f004:**
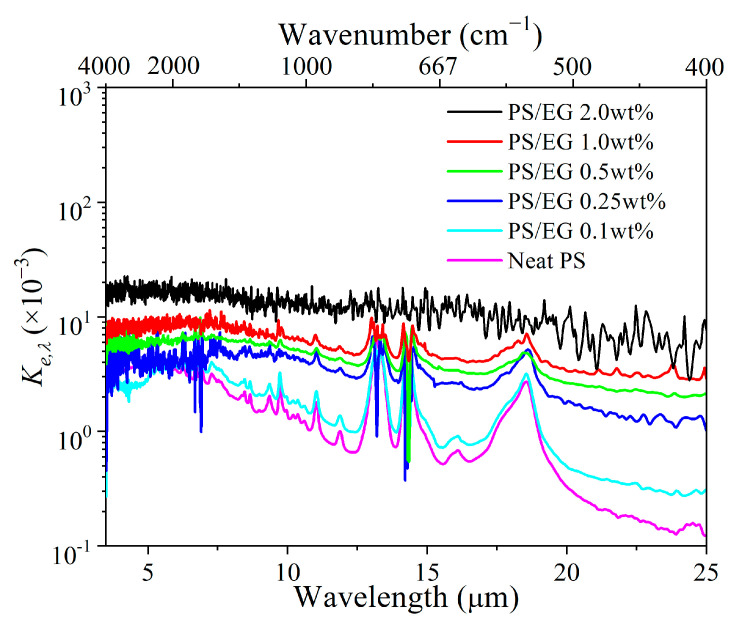
Spectral extinction coefficient of neat PS and PS/EG foams as a function of wavelengths ranging 2.5–25 μm.

**Figure 5 polymers-17-01040-f005:**
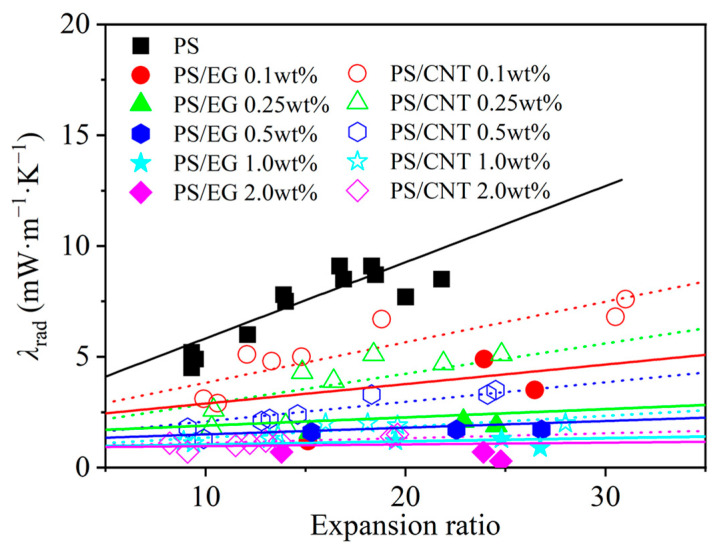
Radiative thermal conductivity in the PS/CNT [5] and PS/EG (linear curve fitting the experimental data) composite foams.

**Figure 6 polymers-17-01040-f006:**
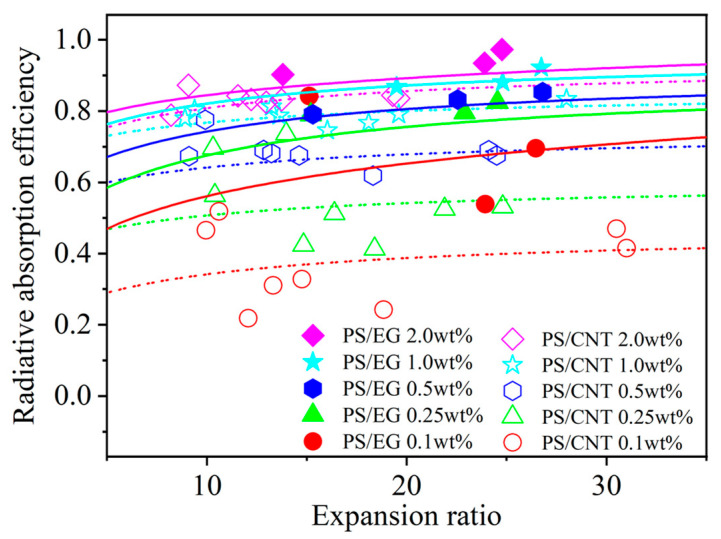
Radiative absorption efficiency of PS/CNT [5] and PS/EG composite foams as a function of the expansion ratio.

**Figure 7 polymers-17-01040-f007:**
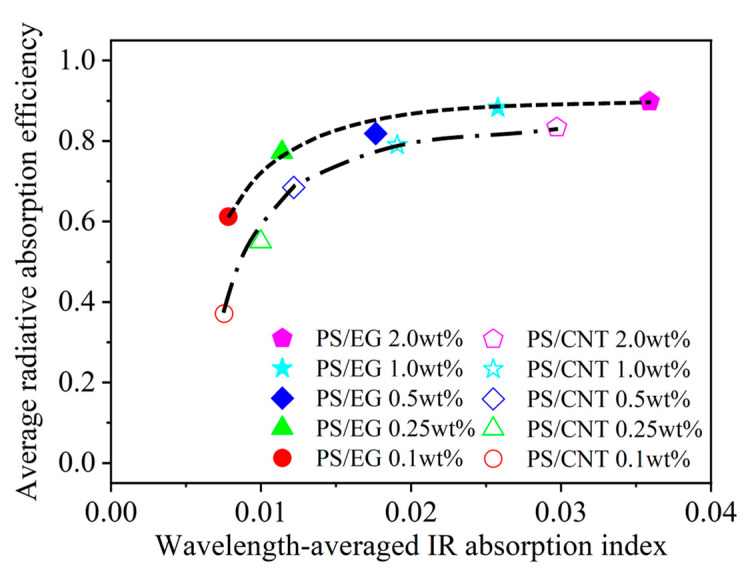
Average radiative absorption efficiency of the PS/CNT [5] and PS/EG composite foams versus the wavelength-averaged IR absorption index of the solid bulk PS/CNT and PS/EG composites.

**Figure 8 polymers-17-01040-f008:**
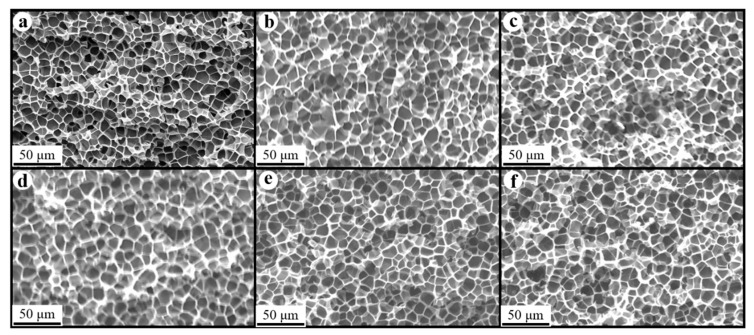
SEM micrographs of PS/EG foams (foamed in scCO_2_–pentane, 13.8 MPa): (**a**) 0 wt% EG; (**b**) 0.1 wt% EG; (**c**) 0.25 wt% EG; (**d**) 0.5 wt% EG; (**e**) 1.0 wt% EG; and (**f**) 2.0 wt% EG.

**Figure 9 polymers-17-01040-f009:**
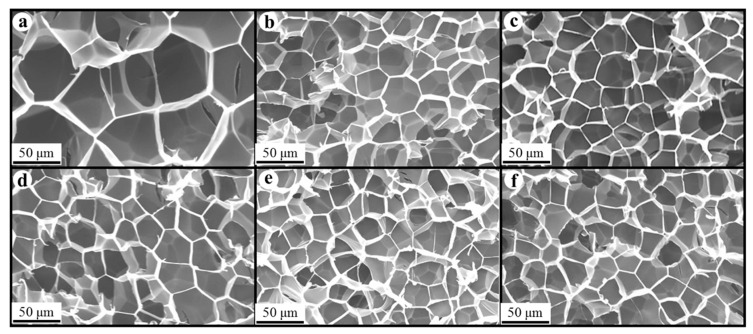
SEM micrographs of PS/EG foams (PS/EG-A) (foamed in scCO_2_–S-propene, 5.9 MPa): (**a**) 0 wt% EG; (**b**) 0.1 wt% EG; (**c**) 0.25 wt% EG; (**d**) 0.5 wt% EG; (**e**) 1.0 wt% EG; and (**f**) 2.0 wt% EG.

**Figure 10 polymers-17-01040-f010:**
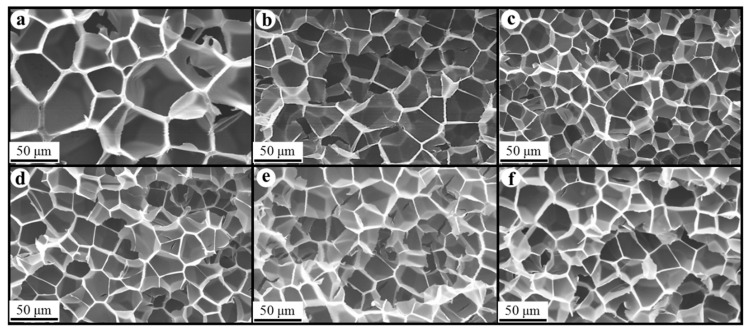
SEM micrographs of PS/EG foams (PS/EG-B) (foamed in scCO_2_–S-propene, 6.6 MPa): (**a**) 0 wt% EG; (**b**) 0.1 wt% EG; (**c**) 0.25 wt% EG; (**d**) 0.5 wt% EG; (**e**) 1.0 wt% EG; and (**f**) 2.0 wt% EG.

**Figure 11 polymers-17-01040-f011:**
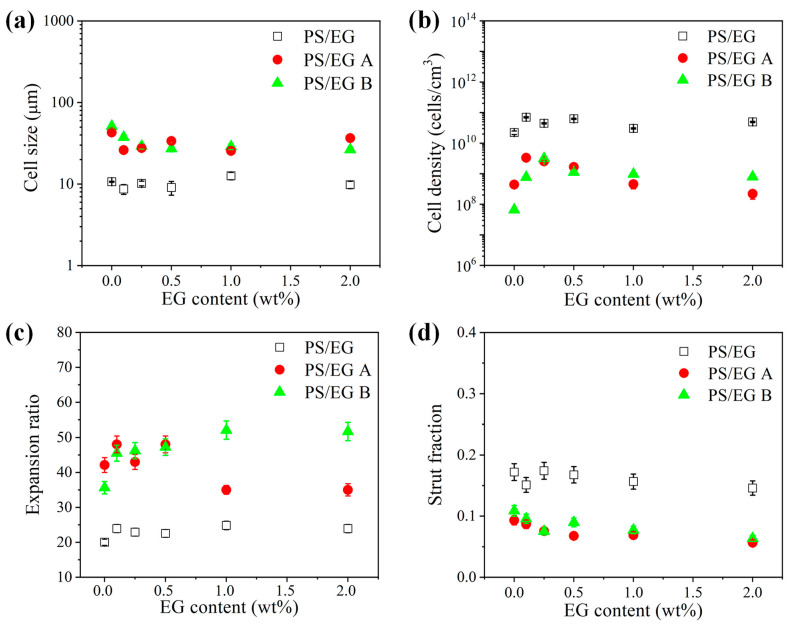
(**a**) Cell size, (**b**) cell density, (**c**) expansion ratio, and (**d**) strut fraction of the neat PS and PS/EG foams as a function of the EG content; PS/EG foamed in scCO_2_–pentane (13.8 MPa), PS/EG-A foamed in scCO_2_–S-propene (5.9 MPa), and PS/EG-B foamed in scCO_2_–S-propene (6.6 MPa).

**Figure 12 polymers-17-01040-f012:**
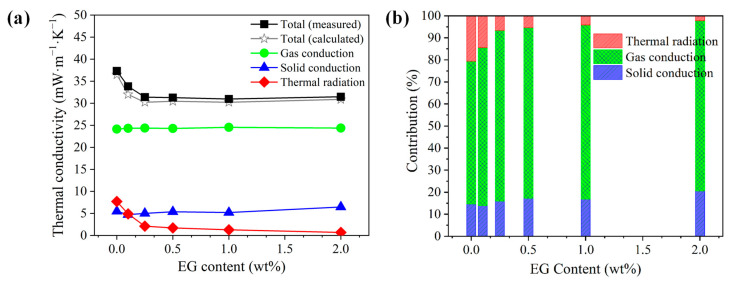
Analyzed thermal conductivity data of the PS/EG samples foamed in scCO_2_–pentane at 13.8 MPa: (**a**) comparison of the experimental and calculated total thermal conductivities and the (calculated) thermal conductivities from gas conduction, solid conduction, and radiation; (**b**) contribution of each heat transfer term in percentage to the total calculated thermal conductivity.

**Figure 13 polymers-17-01040-f013:**
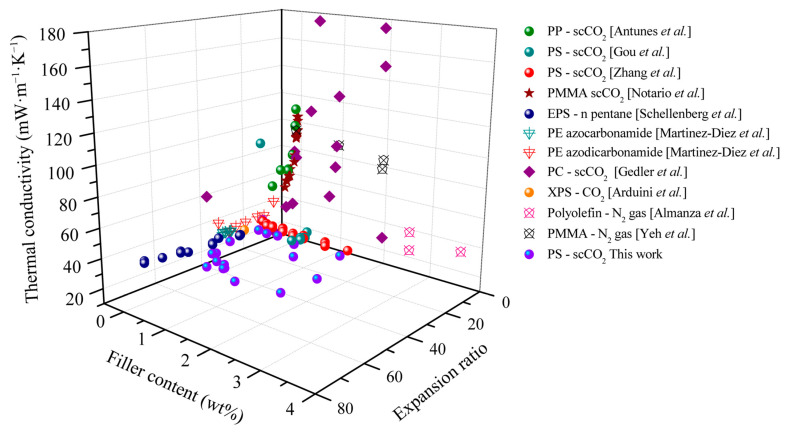
Total thermal conductivity of polymeric foams as a function of the carbonaceous additive content and the expansion ratio. Reference data from PP [Antunes et al.] [45]; PS [Gou et al.] [46]; PS [Zhang et al.] [47]; PMMA [Notario et al.] [48]; EPS [Schellenberg et al.] [49]; PE [Martinez-Diez et al.] [50]; PE [Alvarez-Lainez et al.] [4]; PC [Gedler et al.] [51]; XPS—scCO_2_ [Arduini et al.] [52]; polyolefin—N_2_ gas [Almanza et al.] [14]; PMMA [Yeh et al.] [53].

**Table 1 polymers-17-01040-t001:** Wavelength-averaged IR absorption index of bulk solid neat PS and composites (PS/EG and PS/CNT).

Filler Content (wt%)	Wavelength-Averaged IR Absorption Index (×10^3^)	
	EG	CNT [5]
0.0	4.32	4.32
0.1	7.82	7.54
0.25	11.4	9.97
0.5	17.5	12.2
1.0	25.2	19.0
2.0	35.9	29.7

**Table 2 polymers-17-01040-t002:** Thermal insulation properties of different materials.

Materials	Thermal Conductivity (mW·m^−1^·K^−1^)	References
Silica hollow nanosphere	35	[54]
Nanowood	30	[55]
Polyaniline/pectin biomass	33	[56]
Silk fiber + silica	33	[57]
Resorcinol formaldehyde aerogel	26	[58]

## Data Availability

The original contributions presented in this study are included in the article/Appendix A. Further inquiries can be directed to the corresponding author.

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
