# Peer review of "Thermal Insulation Foam of Polystyrene/Expanded Graphite Composite with Reduced Radiation and Conduction"

_polymers, 2025, doi:10.3390/polym17081040_

Round 1

Reviewer 1 Report

Comments and Suggestions for Authors

After reviewing the provided manuscript titled "Lowering Thermal Conductivity of Polystyrene/Expanded Graphite Composite Foams by Reducing Radiation and Thermal Conduction," here are my observations and comments:

The study addresses a timely topic in material science, focusing on enhancing thermal insulation properties using expanded graphite (EG) in polystyrene foams. The novelty is evident in the systematic analysis of EG's impact and the use of advanced techniques like scCO2-assisted foaming with co-blowing agents. Moreover, the manuscript is generally well-written, but there are instances of grammatical errors, repetition, and formatting issues, especially regarding the citation of figures and tables.

·         The title is descriptive but could be made more concise for better impact.

·         The abstract clearly outlines the study's objectives, methodology, and key findings, but it could benefit from a sharper focus on the practical implications of the results.

·         The introduction could better highlight the environmental benefits of using EG and S-propene for broader appeal.

·         The explanation of the Rosseland equation and its application could be expanded for readers unfamiliar with radiative thermal conductivity calculations.

·         Clarify the rationale for choosing specific EG contents (e.g., 0.1–2.0 wt%) and their alignment with industrial applications.

·         The dependence of the radiative contribution on cell morphology could be elaborated further.

·         Several references to figures are marked as "Error! Reference source not found." Ensure all figures are appropriately referenced and discussed in the text.

  • The conclusion could emphasize the potential industrial scalability and environmental impact of the findings.
  • References are comprehensive and relevant, but ensure consistency in formatting, especially for journal abbreviations and page ranges.
  • Improve language and grammar throughout the manuscript for readability.
Comments on the Quality of English Language

The authors should improve language and grammar throughout the manuscript for readability

Author Response

Thank you very much for your comments. We carefully considered all your
comments and accommodated them to improve the manuscript.

Reviewer 2 Report

Comments and Suggestions for Authors

In this manuscript, the authors have demonstrated a novel processing technique to achieve low thermal conductivity (or superthermal insulating) foams. The unique feature of this work lies in the large expansion ratio, blocking of the IR radiation with apt addition of low cost expanded graphite filler, and a smart use of S-propene co-blowing agent to reduce the gas conductivity. The researchers have provided a clear motivation and background of the paper, as well as added sufficient experimental details and supplementary information for other researchers to replicate the work. They have also provided nice summaries, comparison tables and plots to compare their results with that of the existing literature.

The journal readers' experience would be further enhanced if the authors can clarify the following in their paper:

- Compared to the results obtained from references [42] and [43] shown in Figure 13, add a clear discussion as to why the authors results are much better despite using the same materials.

- In page 8, while discussion Figures 5-7, authors mention that the radiative K increased with increased expansion ratio. Could you explain why this occurs? Is this because of decreased cell wall thickness which minimizes the IR attenuation?

- In Figure 4, the caption for Fig 4(a) is incorrect. It is referred to as extinction coefficient, but the plot does not show it directly. Might help to add the line fit and slope for the referenced values.

- In Figure 4(b), why are the PS related FTIR functional group peaks missing from 2wt% sample? It should still have the same absorption peaks (or transmittance profile). Explain why is the extinction coefficient spectrum flat for 2wt%?

- On page 6, authors stated that "sample thicknesses of PS/EG foams were not consistent but around 500 um." Since film thickness also has a huge impact of IR-transmission, provide an error bar due to this approximation of the film thickness or show thickness normalized Absorbtion.

- In Table 1, add another column showing the foam density value. Without the tabulated value, it is difficult to follow the discussion from later in the text.

- References to the Figures, Tables, and Supporting information figures is not clear throughout the paper. Kindly rectify the labels and clarify which figure in the SI is referred to.

- On page 5, authors mention the average platelet size of EG were <5um. Kindly provide statistical details from the image analysis, how many platelets were counted, and include standard deviation.

- The overall synthesis is a very slow process. For viable manufacturing, how can some of these steps be sped up? Include discussion in the next steps referring to some standard manufacturing techniques.
